



# From a polar to a marine environment: has the changing Arctic led to a shift in aerosol light scattering properties?

Dominic Heslin-Rees[1,2], Maria Burgos[1,2], Hans-Christen Hansson[1,2], Radovan Krejci[1,2], Johan Ström[1,2], Peter Tunved[1,2], and Paul Zieger[1,2]

[1]Department of Environmental Science, Stockholm University, Stockholm, Sweden
[2]Bolin Centre for Climate Research, Stockholm University, Stockholm, Sweden

**Correspondence:** paul.zieger@aces.su.se

**Abstract.**

The study of long-term trends in aerosol optical properties is an important task to understand the underlying aerosol processes influencing the change of climate. The Arctic, as the place where climate change manifests most, is an especially sensitive region of the world. Within this work, we use a unique long-term data record of key aerosol optical properties from

Zeppelin observatory, Svalbard, to ask the question of whether the environmental changes of the last two decades in the Arctic are reflected in the observations. We perform a trend analysis of the measured particle light scattering and backscattering coefficients and the derived scattering Ångström exponent and hemispheric backscattering fraction. In contrast to previous studies, the effect of in-cloud scavenging and potential sampling losses at the site is taken explicitly into account in the trend analysis. The analysis is combined with a back trajectory analysis and satellite-derived sea ice data, to support the interpretation of the

observed trends. We find that the optical properties of aerosol particles have undergone clear and significant changes in the past two decades. The scattering Ångström exponent and the particle light scattering coefficient exhibit statistically significant decreasing of between -4.9 and -6.3 % per year (using wavelengths of $\lambda = 450$ and $550\,\text{nm}$) and increasing trends of between 2.3 and 2.9 % per year (at a wavelength of $\lambda = 550\,\text{nm}$), respectively. The magnitudes of the trends vary depending on the season. These trends indicate a shift to an aerosol dominated more by coarse-mode particles, most likely the result of increases in

the relative amount of sea spray aerosol. We show that changes in air mass circulation patterns, specifically an increase in air masses from the south-west, are responsible for the shift in aerosol optical properties, while the decrease of Arctic sea ice in the last two decades had only a marginal influence on the observed trends.

## 1 Introduction

The Arctic region is warming considerably faster than the global average, a phenomenon known as *Arctic Amplification*.

Svalbard, a Norwegian archipelago located between 74°- 81° N, has experienced some of the greatest observed regional temperature increases throughout the past three decades (Nordli et al., 2014). The impacts of *Arctic Amplification* can be observed in a multitude of parameters (IPCC, 2019), including most notably, reductions in Arctic summer sea ice (Perovich et al., 2018). Diminishing sea ice has been proposed as the leading mechanism to explain the sensitivity of the region in terms of enhanced





sea surface temperature increases (Screen and Simmonds, 2010). However, numerous other mechanisms have been studied

including changes in cloud cover (Schweiger et al., 2008), transportation of heat from the midlatitudes (Overland and Wang, 2016), increases in the total water vapour in the Arctic atmosphere (Park et al., 2015) and sulphate aerosol reductions in Europe (Navarro et al., 2016). The exact relative importance of each of them is debated (Dai et al., 2019; Serreze and Barry, 2011). *Arctic Amplification* may be linked to mid-latitude weather (Cohen et al., 2014; Pithan et al., 2018), with increased Rossby wave amplitude (Francis et al., 2017) and the appearance of atmospheric circulation anomalies (Lee et al., 2015). Changes

in air circulation patterns within the Svalbard region have been observed and linked to changes in the Arctic Oscillation; the changes include increased anticyclonic advection from the south during winter and from the north during summer (Maturilli and Kayser, 2017).

Aerosol exert a considerable influence on the Arctic climate (Willis et al., 2018). The IPCC (2013) report that in combination with clouds, aerosols continue to contribute the largest uncertainty to our understanding of changes to the Earth's energy

budget. Aerosols can influence the Earth's climate directly through aerosol-radiation interactions. Arctic aerosol have a net cooling effect on the region (Fyfe et al., 2013), whereby aerosol particles reflect shortwave radiation. With the significant decrease in sulphate aerosol in the past few decades (Quinn et al., 2007), research into aerosol pollution in the Arctic has been gaining renewed interest as a means of explaining the increased rate of warming within the Arctic region (Garrett and Verzella, 2008; Navarro et al., 2016). The small amount of solar radiation that the Arctic receives means the region is fairly sensitive to

any perturbations in radiative fluxes (Valero et al., 1989). In addition, aerosol particles may act as a layer of light absorbing material over a highly reflective ice/snow surface (Sokolik et al., 2010). Changes in aerosol concentrations are also likely to have an impact on cloud condensation nuclei (CCN) concentrations, given the two are well correlated (Jung et al., 2018). The strong seasonal cycles in Arctic aerosol properties have been well documented (Ström et al., 2003; Tunved et al., 2013; Freud et al., 2017; Pandolfi et al., 2018; Schmeisser et al., 2018). Observations from in situ ground-based sites have helped

to describe this annual cycle. The Zeppelin Observatory (ZEP), Svalbard experiences a typical Arctic annual cycle; there are peaks in accumulation-mode concentrations in late spring and early winter (Freud et al., 2017) in accordance with the Arctic haze phenomenon. The appearance of the Arctic Haze phenomenon is a result of the combined effects from the increased south-to-north transportation of aerosol and a decline in removal efficiency during northward transport (Shaw, 1995; Quinn et al., 2007). Summer, with more frequent cloudiness and low-intensity precipitation events within a retracted Polar Front, is

nearly free of anthropogenic aerosol influence (Willis et al., 2018) with the exception of well-defined events related to biomass burning and forest fires (Warneke et al., 2009). Summertime experiences increased concentrations of smaller nucleation and Aiken-mode particles, formed in situ (Freud et al., 2017). The light scattering properties at ZEP have been shown to reflect these changes in aerosol composition, with a scattering peak in the late winter and early spring (Schmeisser et al., 2018; Pandolfi et al., 2018; Heintzenberg, 1982). In addition, optical properties at ZEP signal an aerosol composition composed of smaller

aerosol in the spring and larger aerosol in the late summer. More details on the seasonality of in situ optical properties can be found in Pandolfi et al. (2018) and Schmeisser et al. (2018).

This study focuses on key aerosol optical properties needed to describe the aerosol interaction with solar radiation; these properties are needed to accurately estimate the direct aerosol radiative forcing (Haywood and Shine, 1995). We refer to





both extensive and intensive aerosol optical properties; the intensive parameters are calculated from extensive parameters and

thus independent of the amount of aerosol. The extensive parameters include scattering coefficients ($\sigma_{\mathrm{sp}}$) and backscattering coefficients ($\sigma_{\mathrm{bsp}}$), whilst the intensive parameters are the scattering Ångström exponent ($\alpha$) and the hemispheric backscattering fraction ($b$). $\lambda$ is omitted from the abbreviations, in subsequent sections, for simplicity, however, it should be noted that the above optical properties are dependent on $\lambda$.

The hemispheric backscattering fraction $b$ is defined as

$$b(\lambda) = \frac{\sigma_{\mathrm{bsp}}(\lambda)}{\sigma_{\mathrm{sp}}(\lambda)} \tag{1}$$

where $\lambda$ denotes the wavelength, $\sigma_{\mathrm{bsp}}$ is the hemispheric backscattering coefficient and $\sigma_{\mathrm{sp}}$ the particle light scattering coefficient.

The scattering Ångström exponent $\alpha$ describes the wavelength dependency of the particle light scattering coefficient and is defined as

$$\alpha_{\mathrm{sp}} = -\frac{\ln\left(\sigma_{\mathrm{sp},1}/\sigma_{\mathrm{sp},2}\right)}{\ln\left(\lambda_1/\lambda_2\right)} \tag{2}$$

where $\sigma_{\mathrm{sp},1}$ and $\sigma_{\mathrm{sp},2}$ are the particle light scattering coefficients at wavelengths $\lambda_1$ and $\lambda_2$.

$\alpha$ represents the wavelength dependency of particle light scattering. $\alpha$ is inversely proportional to the size of the aerosol, thus larger particles exhibit lower $\alpha$ values and vice versa.

Long-term measurements are vital to understanding changes related to the Arctic, and thus to the Earth's climate. There

is a need for long-term measurements to help reduce the uncertainties surrounding the impacts of aerosol (Hansen et al., 1996). Ground-based in situ measurement sites offer the chance to examine the long-term trends associated with aerosol optical properties. However, there are numerous difficulties in maintaining continuous and long-term observations, leading to a shortage of multidecadal time series. It is, therefore, essential to use the available data sets to examine trends, and research the underlying mechanisms behind changes in aerosol properties. Aerosol optical properties experience a high degree of natural

variability at different temporal resolutions and therefore pose difficulties in determining the underlying long-term trends. Sites in polar regions exhibit the largest number of statistically significant positive trends in particle light scattering (Collaud Coen et al., 2013, 2020), with multiple Arctic sites experiencing alternating trend slopes based on the duration of the trend (most notably in Barrow, Alaska). Interestingly, most Arctic sites do not reflect the overall decreasing trends in scattering coefficients observed throughout Europe and North America (Collaud Coen et al., 2020). Thus, it is important to study why these differences

are present and the mechanisms underlying these trends. Exploring the reasons for these trends is a central topic of this study, and separates it from most previous long-term trend analyses. This study further differs from Collaud Coen et al. (2020) in that a longer data set is used and that the data is pre-screened with respect to ambient relative humidity. The trends in aerosol measurements cited in this study are all conducted under dry conditions, as opposed to ambient conditions. Dry conditions, where the relative humidity (RH) is controlled, help to minimise the effects of water uptake, known as hygroscopicity. It should

be noted that the hygroscopic effect on particle light scattering is more pronounced in the Arctic, compared with measurements taken in other global regions (Zieger et al., 2010, 2013).



This study aims to show long-term trends in Arctic aerosol optical properties, in combination with back trajectory analysis and satellite-derived sea ice data, to help better understand the changing processes controlling the optical properties of Arctic aerosol. Connecting back trajectory data with aerosol optical measurements helps to assess the potential aerosol sources and types. Not only does this study explore long-term trends in aerosol optical data, but it provides an analysis of the mechanisms influencing these changes. The influence on Arctic aerosol from sea ice retreat remains poorly constrained (Willis et al., 2018). One particular motivation for this study concerns the retreat in Arctic summer sea ice, which has seen significant reductions throughout the same period. The retreat of Arctic sea ice has the potential to induce changes to the aerosol composition; model results suggest an increase in sea salt aerosol (SSA) emissions from the loss of summer ice (Struthers et al., 2011; Browse et al., 2014). The aims of this study are to answer the following two research questions: (a) At what rate have key aerosol optical properties changed at Svalbard, in the Arctic, during the last two decades? (b) Can changes in long-term aerosol optical properties be explained by meteorological parameters and/or changes in sea ice coverage?

## 2 Materials and Methods

### 2.1 Measurement site

The study is conducted using aerosol optical data from Zeppelin Observatory (ZEP), located at 78.91°N, 11.89°E at an altitude of 474 m a.s.l. on the western edge of the Norwegian Archipelago, Svalbard. The station is part of the World Meteorological Organisation's (WMO) Global Atmosphere Watch (GAW) programme. The nearby research village Ny-Ålesund is located on the coast of Kongsfjorden, approximately 2 km away from ZEP. The remoteness and altitude of the observatory allow for it to measure pristine Arctic air with minimal contamination from local pollution, whilst also being able to observe long-range transported pollution. The location of ZEP as an Arctic site is unique, given its proximity to European sources and closeness to the Arctic Ocean sea ice edge. The nearby sea at ZEP is open all year round. Due to its elevated location, its exposure to free-tropospheric air and cloudiness separates it from other Arctic sites (Freud et al., 2017).

### 2.2 Nephelometer

At ZEP, measurements of aerosol light scattering properties are performed using an integrating nephelometer (TSI Inc., U.S.A., Model 3563) since May 1999, making it, next to Barrow (Alaska) and Alert (Canada), one of the longest time series records in the Arctic. The nephelometer performs continuous measurements of the light scattering of aerosol particles at three wavelengths ($\lambda$ = 450, 550, 700 nm). The particle light scattering, $\sigma_{\mathrm{sp}}(\lambda)$ (m$^{-1}$) and the hemispheric backscattering $\sigma_{\mathrm{bsp}}(\lambda)$ (m$^{-1}$) coefficients are recorded, without the need to know any information about size, composition, and the physical state of the light-scattering aerosol. RH, temperature, and pressure sensors also provide accurate readings at either the sample inlet or outlet. The nephelometer is regularly calibrated using $CO_2$ and particle-free air. The contribution of light scattering by air molecules (Rayleigh scattering) is automatically corrected for by regular zero measurements of particle free air, about every hour. Gaps in the data set are present, due to either instrumental failure, in which the measurements did not perform correctly or the instru-





ment was away for servicing. The nephelometer is connected to a whole-air inlet which follows the guidelines of WMO/GAW
for aerosol sampling (WMO, 2016) with similar characteristics as the inlet described by Weingartner et al. (1999). Due to the

temperature difference between inside and outside, and the additional heating of the inlet, no additional drying of the aerosol
is needed. The aerosol is sampled at dry conditions with RH = $7.0 \pm 4.9$ % (mean $\pm$ standard deviation; SD) to maintain the
GAW recommended conditions of RH < 40 %.

The observatory, housing the instrumentation, was demolished in 1999 and rebuilt and reopened at the same location in
May 2000. The nephelometer itself has undergone repairs and a change to its inlet was made in June 2011. Detailed logbooks

during the initial years of the observatory are lacking, providing difficulties in the traceability of past operating procedures
and calibration. However, the manual inspection of all data and removal of periods with the obvious malfunctioning of the
instrument is performed. More details on the nephelometer data treatment are given below.

### 2.3  Back trajectory analysis

Air mass back trajectories are calculated every hour, with the air parcels arriving at the altitude of ZEP. The HYSPLIT model

(Draxler and Hess, 1998) is used to perform the back trajectory calculations. The meteorological fields are obtained from
NOAA; the period 1999-2004 uses the FNL achieve, and 2005-2016 uses the Global Data Assimilation System (GDAS)
(http://ready.arl.noaa.gov/archives.php).

### 2.4  Data treatment

The hourly medians of aerosol optical properties, back trajectory, and ambient relative humidity data are temporally collocated.

Approximately, ~59.2 % of the hourly aerosol measurements are left in after the quality control procedure and temporal
collocation of the data set. The period of study is from 1999 to the end of 2016.

This section concerns the data processing of the following:

1. Treatment of optical properties: cleaning and computing medians for $\sigma_{sp}$, $\sigma_{bsp}$, $\alpha$, and $b$.

2. Use of station meteorology data.

3. Back trajectory calculations: combining land type data and determining source regions.

#### 2.4.1  Treatment of optical properties

The data is quality checked using similar procedures to Asmi et al. (2013), whereby if a change in the instrumental conditions
is coincident with a clear change in the optical properties, the data of the changed period is not included. Most notably, obvious
outliers are removed. When available, the information from logbooks is used to remove periods of known instrument failure.

Data are selected such that the RH at the sample outlet never surpasses 40 %, this helps minimise the effects on $\sigma_{sp}$ and $\sigma_{bsp}$ from
hygroscopic growth (see e.g., WMO, 2016). Hourly medians, composed of at least 5 data points, are calculated (see Fig. S1 in
the supplement). The particle light scattering coefficients are adjusted to standard temperature and pressure, using data from the





in-built instrument sensors. Detection limits (Anderson et al., 1996) and the illumination and truncation error correction method by Anderson and Ogren (1998) are applied to the hourly medians, based on their respective wavelength and scattering type.

The detection limits for $\sigma_{sp}$ and $\sigma_{bsp}$ ($\lambda = 550$ nm) are 0.37 and 0.24 Mm$^{-1}$ respectively and represent the linearly-interpolated estimations for a one hour averaging time, based on Anderson et al. (1996). Small values for $\sigma_{sp}$ are considered less reliable due to instrument noise at low aerosol loadings (Schmeisser et al., 2017), and studies often use a constant, Schmeisser et al. (e.g., 2018) consider $\sigma_{sp} > 1$. However, for this study, thresholds on particle light scattering coefficients are applied based on detection limits, as to not bias the data and push the extensive values higher. Overall, the fraction of data removed in terms

hourly averages are as follows: 33.5% for $\sigma_{sp}$ ($\lambda = 450$ nm), 28.7% $\sigma_{sp}$ ($\lambda = 550$ nm) and 41.3% $\sigma_{bsp}$ ($\lambda = 550$ nm). Most of the years are not affected by missing data with the exception of the years 2003 and 2016, where 67% and 67%-74% of data is excluded respectively. From these data, Eqs. 1 and 2 are used to calculate $b$ and $\alpha$, respectively. $b$ is calculated based on the green (550 nm) wavelength, and $\alpha$ is computed from the blue (450 nm) and the green wavelengths, as the red channel of the nephelometer generally exhibits greater variability. However, it should be noted that the results do not change significantly if a

fit over all three channels is being used.

Daily and seasonal medians are computed and used to assess the trends in aerosol optical properties. Daily medians are based on a minimum of 6 hourly data points (in keeping with a 25% threshold imposed by Collaud Coen et al. (2020)). Daily medians reduce the noise related to very low aerosol loadings; any uncertainties in $\sigma_{sp}$ and $\sigma_{bsp}$ are more pronounced in the Arctic region and are further enhanced in the derived intensive properties, $b$ and $\alpha$.

### 170 2.4.2 Relative humidity

When the observatory is in the midst of a cloud, large water droplets taken in by the whole-air inlet may affect the collection efficiency of the inlet. In addition, particles in the surrounding air can be removed due to in-cloud scavenging. This study aims to take this into account, however, for heated whole-air inlets, inside-cloud situations should not affect the measurements significantly (Asmi et al., 2013). Hourly mean values of ambient RH are used as a proxy for the presence of clouds. The

ambient RH for ZEP is acquired through EBAS (http://ebas.nilu.no/). The ambient RH measurements are operated by the Norwegian Institute for Air Research, NILU, Atmosphere and Climate Department. The data is pre-screened to remove periods where ambient RH exceeded 95 %, where it is assumed that cloud conditions were present for some fraction of that hour. The measurements are therefore assumed to be of cloud-free conditions (see Fig. S2 in the supplement). The chosen threshold is the point at which a significant drop in $\sigma_{sp}$ is observed. The observatory has frequent inside-cloud situations, which can affect

the aerosol optical measurements; approximately 10 % of the optical data is removed as a result of high ambient RH values, with summer the most affected season, $\sim 20.7\%$ is removed, as opposed to $\sim 5.9\%$, $\sim 12.4\%$, $\sim 6.6\%$ for spring, autumn, and winter, respectively.

### 2.4.3 Trajectory calculations

Each back trajectory is 7 days in length; the number of days is chosen as a compromise between restricting the increasing

uncertainties the further back in time they go, and capturing the typical lifetime of the aerosol in the atmosphere, and with




this the main source regions. The HYSPLIT back trajectories provide a detailed history of the air parcels, including changes in latitude, longitude, altitude, the height of the mixed layer (ML), precipitation, relative humidity and temperature. The ML forms part of the planetary boundary layer (PBL); it is defined as the height at which substances, such as aerosol, can be vertically dispersed and well-mixed (Seinfeld and Pandis, 2006). The back trajectories are classified as being above or below

the ML height. In the interest of this study, only data points (time) where the air parcels reside within the ML are considered to be influenced by surface sources and dominate the observed signal at ZEP.

Wind speed is a main driver to SSA production and thus highly pertinent to this study. The average wind speeds are calculated as part of the back trajectory analysis based on the change in latitude, longitude, and altitude of each respective hourly data point. The direction from which back trajectories arrive at ZEP is computed by calculating the mean Cartesian-transformed

coordinates. The back trajectories are separated into four regions namely, north-west, north-east, south-east, and south-west. The four regions are defined relative to ZEP being at the centre. Monthly Special Sensor Microwave/Imagers satellite-derived sea ice concentration (SIC) data from the Version 1.1 Hadley Centre Sea Ice (HadISST1.1) dataset (Met Office, 2006) is used. The SIC data set is of a 1° x 1° resolution. SIC data are temporally collocated with the back trajectories. Each data point from the back trajectory is classified into the following: (1) above ML, (2) above land and within ML, (3) above sea ice and within

ML, and (4) above open water and within ML. The ice surface type is defined as having a mean SIC above 0.85 (see Fig. S3 in supplement) which is commonly used in the literature (e.g., Rayner et al., 2003; Fetterer et al., 2016; Stroeve and Meier, 2003) although lower thresholds have been used as well.

## 2.5 Statistical tools for trend analysis

For all the trend analyses, the Python *scipy.stats* package is used within *SciPy* (v.1.1.0) (Jones et al., 2001–).

### 2.5.1 Seasonal Mann-Kendall test

The Mann Kendall test (hereafter, MK test) is one of the most widely used non-parametric tests for determining the statistical significance of trends in environmental data. The test is rank-based with a null hypothesis that the time series has no trend. The alternative hypothesis is that a monotonic trend exists (Gilbert, 1987). The MK test is ideal since it is not affected by missing values. Given that the data set contains gaps and irregular spacing, this is a major advantage. Hirsch et al. (1982) developed

the Seasonal MK test to take seasonality into account. The Seasonal MK test works by separating the data into distinct seasons and then comparing like for like.

The MK test assumes that the data are independent, thus there is no autocorrelation. Kulkarni and von Storch (1995) show that the result of the MK test depends strongly on the autocorrelation; a positive autocorrelation increases the likelihood that the null hypothesis is rejected when there is no trend. The time series is pre-whitened (PW) based on the trend-free pre-whitening

(TFPW) procedure described in detail in Yue et al. (2002).

The daily medians provide a sufficient amount of data points, however, are considerably more plagued by autocorrelation than time series constructed from longer averaging periods (i.e. seasonal medians). The seasonal MK test is performed on





the PW daily medians with a two-tailed significance test of 95 %. Medians are used for all the analyses as the variables are non-normally distributed. Medians are also less affected by outliers.

### 2.5.2 The Theil–Sen estimator

The Theil–Sen estimator (TS) is used in this study to calculate the slope of the linear trends. The procedure developed by Sen (1968) is useful, as it is relatively less affected by outliers. TS is a non-parametric tool (therefore no type of distribution needs to be assumed for the data) and often accompanies the MK test. TS works by calculating the median of all the slopes of every pair of ordered data points. The slope is unbiased in regards to highly autocorrelated data, however standard errors are affected by autocorrelation. The estimated slopes of the regression lines are multiplied by 365.25 or 4 for daily and seasonal medians respectively. The relative trends (i.e. % $yr^{-1}$) are obtained by dividing by the median of the data set in question, similar to previous studies (e.g., Asmi et al., 2013).

### 2.5.3 Least Mean Square analysis (LMS)

The Least Mean Square analysis (LMS) is applied to the logarithm of the dependent variable to calculate the relative trends. The logarithm approximates the data to a normal distribution, allowing for the use of this statistical tool.

$$\log(X_t + s) = a + \beta \cdot t + \epsilon_t \tag{3}$$

where: $X_t$ is the observation at time $t$; $a$ is the regression parameter representing the intercept; $\beta$ is the regression parameter representing the slope, while $s$ is a small scalar. $\epsilon_t$ is the random error term associated with each observation.

The confidence level is set at 95 % and the trend is deemed statistically significant (ss) if $|\beta/\mathrm{se}_\beta| > 1.960$, where $\mathrm{se}_\beta$ denotes the standard error of the slope, $\beta$. For LMS trends not in terms of relative changes, the logarithm is not applied to $X_t$.

## 3 Results

Section 3.1 focuses on long-term trends, in which trends in aerosol light scattering properties are presented in Sect. 3.1.1, and the trends in key meteorological parameters from the back trajectory analysis and the changes in source types and regions are presented in Sect. 3.1.2. The trends are calculated and presented based on both seasonal and daily medians. The seasonality of the trends is also included. The final part of the results section, Sect. 3.2, combines the aerosol and back trajectory parameters.

### 3.1 Trend analysis

### 3.1.1 Aerosol optical properties

The long-term trends in $\sigma_{sp}$, $\sigma_{bsp}$, $b$, and $\alpha$ are presented in Fig. 1 based on the seasonal medians. $\sigma_{sp}$ and $\sigma_{bsp}$ both display increasing statistically significant trends estimated to be in the range of 0.05 and 0.01 $Mm^{-1}yr^{-1}$ respectively. $\sigma_{sp}$ and $\sigma_{bsp}$ show clear seasonality with higher scattering coefficients occurring in spring and winter. $b$ displays a non-statistically significant





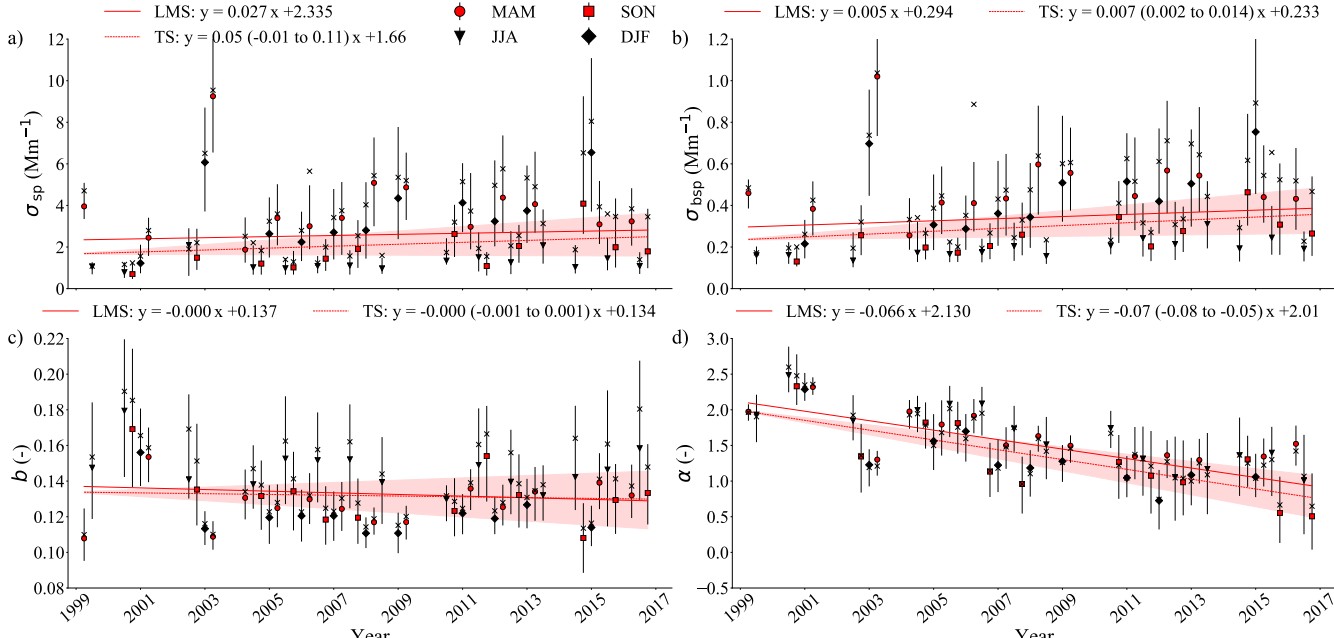

**Figure 1.** Long-term trends of the seasonal medians for a) the particle light scattering coefficient ($\lambda = 550\,\text{nm}$) b) the particle light backscattering coefficient ($\lambda = 550\,\text{nm}$) c) the hemispheric backscattering fraction ($\lambda = 550\,\text{nm}$) d) the scattering Ångström exponent ($\lambda_1 = 450\,\text{nm}$, $\lambda_2 = 550\,\text{nm}$). The seasonal medians are denoted by their respective symbols. The error bars denote the length of the 25th and 75th percentile values. The seasonal mean is given by the cross. The solid and dashed red lines represent the least mean square (LMS) and Theil-Sen slope (TS) of the seasonal medians, respectively. The red shaded area denotes the associated 90 % confidence interval of the TS slope. Note that TS is not used to test the statistical significance.

decreasing trend of -0.0002 $\text{yr}^{-1}$. $\alpha$, however, shows a large and decreasing statistically significant trend of approximately -0.07 $\text{yr}^{-1}$. $\alpha$ is largest during the spring and summer, whilst autumn experiences the smallest medians. $b$ experiences a peak in summer and is at its lowest value in winter. The end of 2002 and the start of 2003 display relatively increased $\sigma_{\text{sp}}$ and $\sigma_{\text{bsp}}$ seasonal medians compared with medians either side. The seasons in question exhibit a decreased $\alpha$ value. Tunved et al.

(2013) observed a reduced aerosol volume size distribution in 2002, which may explain the low $\alpha$ medians for these seasons. The trends based on seasonal and daily medians are displayed in the supplement (see Tables S1, - S4). It should be noted that the trends calculated based on daily medians exhibit reduced magnitudes. The removal of inside-cloud situations had a negligible effect on the overall trends (see Figs. S2 in supplementary material).

The trends for each season, using the MK and LMS methods are examined in Fig. 2. The level of significance is set at 95%

for both methods. The two statistical methods agree well with one another, except for $\sigma_{\text{bsp}}$ and $b$ where the LMS method presents reduced magnitudes for the trends; the increased frequency of low values amongst $\sigma_{\text{bsp}}$ and $b$ is a likely explanation for the inconsistency. $\sigma_{\text{sp}}$ and $\sigma_{\text{bsp}}$ exhibit statistically significant increasing relative trends for all seasons except spring, with the most prominent trends during winter ($\sim 6.0\,\text{to}\,6.1$ and $\sim 1.9\,\text{to}\,5.3\,\%\text{yr}^{-1}$ for $\sigma_{\text{sp}}$ and $\sigma_{\text{bsp}}$ respectively). The overall





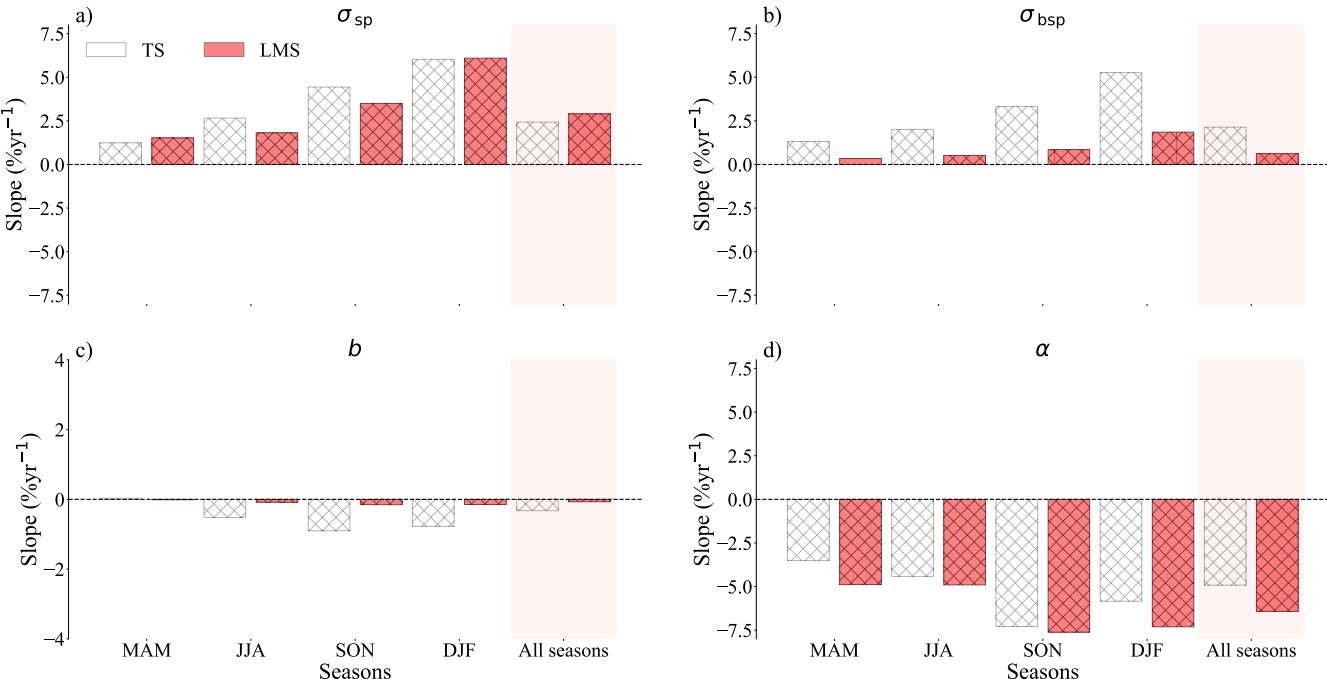

**Figure 2.** Relative trends based on daily medians for a) particle light scattering coefficient, b) particle light backscattering coefficient, c) hemispheric backscattering fraction (note the different y-scale), and d) scattering Ångström exponent for different and all seasons. The white bar displays the Theil-Sen estimator (TS). The red bar displays log-transformed Least Mean Square (LMS) trends. Crosshatching denotes trends that are statistically significant (ss) at a confidence interval of 95 %. The ss for the TS is based on "prewhitened" (PW) time series. The trends in their respective units $\mathrm{yr}^{-1}$ are in the tables in the appendix.

trend in $\sigma_{\mathrm{sp}}$ is $\sim 2.4$ to $2.9$ $\%\,\mathrm{yr}^{-1}$. $\sigma_{\mathrm{bsp}}$ displays a trend of $\sim 0.6$ to $2.1\,\%\,\mathrm{yr}^{-1}$. $\sigma_{\mathrm{sp}}$ and $\sigma_{\mathrm{bsp}}$ display a similar pattern in the

magnitudes of the different seasons. $b$ displays a slight statistically significant decreasing trend during for all seasons, except spring. The largest decreasing trends are in autumn (-0.2 to -0.9 $\%\,\mathrm{yr}^{-1}$) and winter (-0.2 to -0.8 $\%\,\mathrm{yr}^{-1}$). $\alpha$, however, exhibits statistically significant decreasing relative trends for all seasons. The most prominent seasonal trend for $\alpha$ is during autumn (-7.3 to -7.6 $\%\,\mathrm{yr}^{-1}$). Autumn and summer are the cleanest periods of the year at ZEP and despite low concentrations during these periods, these seasons display the strongest trends in $\alpha$. The overall trend in $b$ is statistically significant and decreasing

based, $\sim 0.07$ to -0.3 $\%\,\mathrm{yr}^{-1}$. $\alpha$ displays an overall trend of $\sim$ -4.9 to -6.4 $\%\,\mathrm{yr}^{-1}$.

### 3.1.2    Trajectory analysis

The relative accumulated times back trajectories spend above each surface type are shown in Fig. 3a. The back trajectories spend approximately 34 % of the time within the ML, however, this undergoes significant interannual variability. The relative contributions for open water, land and ice are 17%, 5%, 12% respectively. The contribution from open water increases over

time and is much more apparent in the last few years. The contribution from land, by comparison, shows little long-term



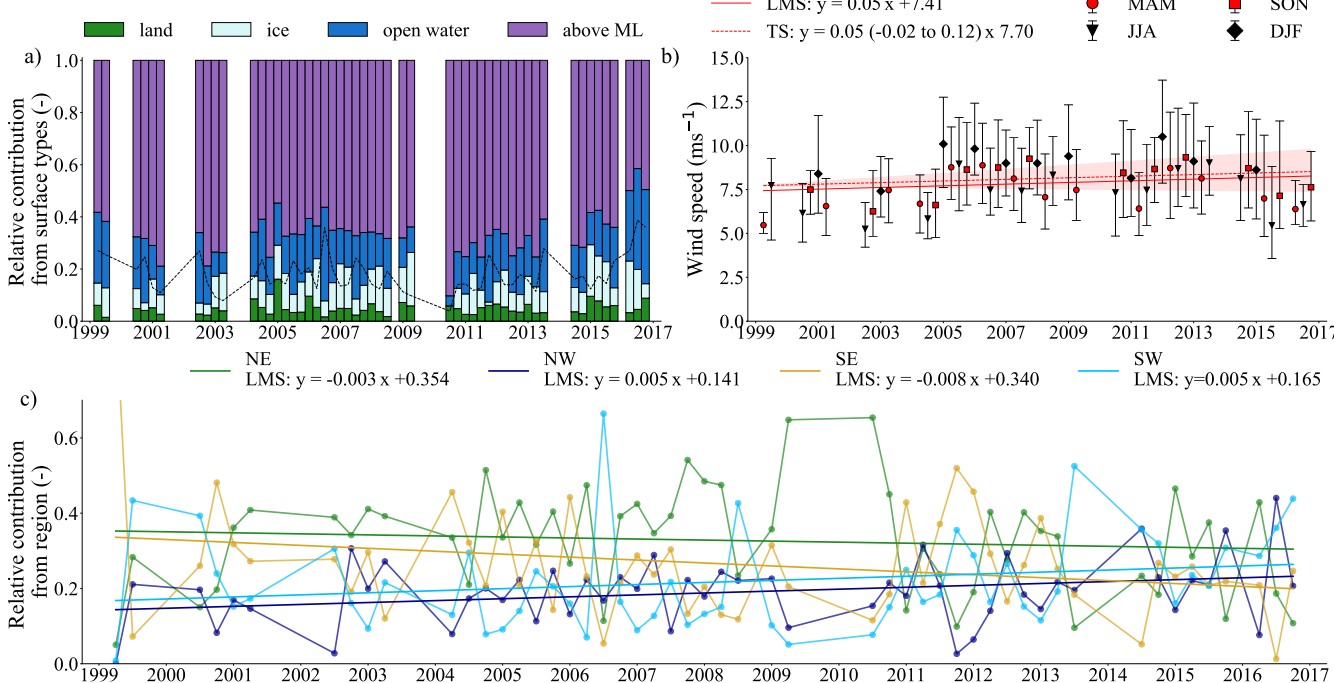

**Figure 3.** Long-term trends in a) relative contribution of the accumulated time per season air parcels spent above land, ice and, open water (whilst within the ML), and above the ML. b) median wind speed of air parcels located within the ML c) relative contribution from the different four regions. The dashed black line in a) presents the proportion of accumulated seasonal times for open water. For b), the dashed red line represents the Theil-Sen slope (TS) of the seasonal medians. The error bars denote the length of the 25th and 75th percentile values. The red shaded area denotes the associated 90 % confidence interval of the TS slope. The solid lines in b) and c) display the Least Mean Square (LMS) regression fit. Note the scale for the y-axis of c) is reduced to 0.6 and south-east contribution in spring 1999 is ~94%.

change throughout. The seasonal variability in the surface types is displayed; the winter and spring seasons are slightly more prominent for land, reflecting the expansion of the Polar Front. Furthermore, the time above ice is noticeably more reduced during summer and autumn months. Figure 3b shows that the median wind speed of air parcels along the back trajectories and within the ML has increased slightly. The wind speeds have undergone an absolute increase of ~0.83 ms$^{-1}$ throughout

the period in question. It is further noticeable that air parcels experience greater wind speeds during the winter, whilst the summer displays a reduction in wind speeds; this is in keeping with the retreat and advance of the Polar Front, allowing back trajectories to travel further. Figure 3c details the relative contributions from the four specified regions namely the NE, NW, SE, and SW. In terms of changes in air circulation patterns, the back trajectories from the SW, most likely crossing over the Atlantic Ocean, make up an increasing proportion of the air masses arriving at ZEP. Both regions to the west have observed an

increase in their relative contribution, whereas air masses predominantly arriving from the east have witnessed a reduction; the NE and SE made up the majority of air mass contributions (i.e. ~34 % and 35 %, respectively) at the start of the study period,





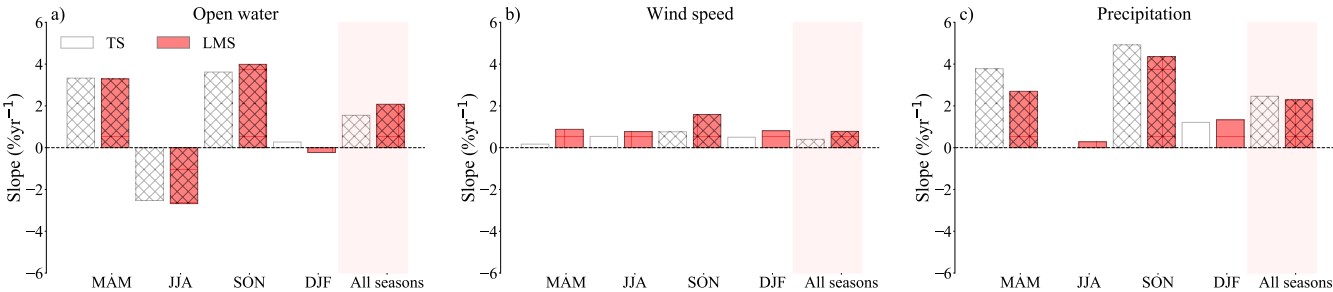

**Figure 4.** Relative trends in daily medians for a) time spent above open water and within the ML, b) median wind speed, c) accumulated precipitation along each back trajectory for different and all seasons. The white bar displays the Theil-Sen estimator (TS). The red bar displays log-transformed Least Mean Square (LMS) trends. Crosshatching denotes trends that are statistically significant (ss) at a confidence interval of 95%. The ss for the TS is based on "prewhitened" (PW) time series.

however, the years towards the end of study period exhibit a more equal mix of air mass origins. It is noticeable that peaks in the relative contribution from open water by season (signified by the black dashed line in Fig. 3a) coincide with seasons experiencing greater contributions from the SW.

The relative trends, in terms of percentage increase, for back trajectory parameters based on daily medians are shown in Fig. 4. The time spent over open water and within the ML displays large statistically significant trends, in particular, spring and autumn show large positive relative changes ($3.3\,\%\mathrm{yr}^{-1}$ and $3.6\,\text{-}\,4\%\mathrm{yr}^{-1}$ respectively), whilst spring shows a decreasing trend. The trends in wind speed are statistically significant for autumn and across all seasons, but $< 1\,\%\mathrm{yr}^{-1}$. The trends in daily medians for open water and precipitation display similar patterns in their respective magnitudes amongst the different
seasons.

Figure 5 displays the relative contributions from the four regions namely the NE, NW, SE, and SW. Note that these relative trends are computed based on monthly contributions, due to the high presence of zero values in the daily values. The relative trends for air masses arriving from the west are noticeably large in magnitude, in particular the south-west. The south-west displays statistically significant increasing trends for autumn and across all seasons ($5.6\,\text{-}\,6.6\%\mathrm{yr}^{-1}$ and $3.4\,\text{-}3.5\%\mathrm{yr}^{-1}$ re-
spectively). The relative trends are decreasing for both the air masses arriving from the east. The north-east displays a large statistically significant decreasing trend in autumn. The relative trends for time spent over the open water (Fig. 4a) and the contribution from SW air masses (Fig. 5c) also display similar variations in the magnitudes for each season.

## 3.2  Increased contribution from coarse-mode particles

$\alpha$ is the only aerosol light scattering property featured in this part of the results as it shows the largest change of the last
two decades. $\alpha$ is a qualitative indicator of aerosol particle size (Ångström, 1929), where $\alpha \leq 1$ indicates an aerosol size distribution dominated by the coarse mode.





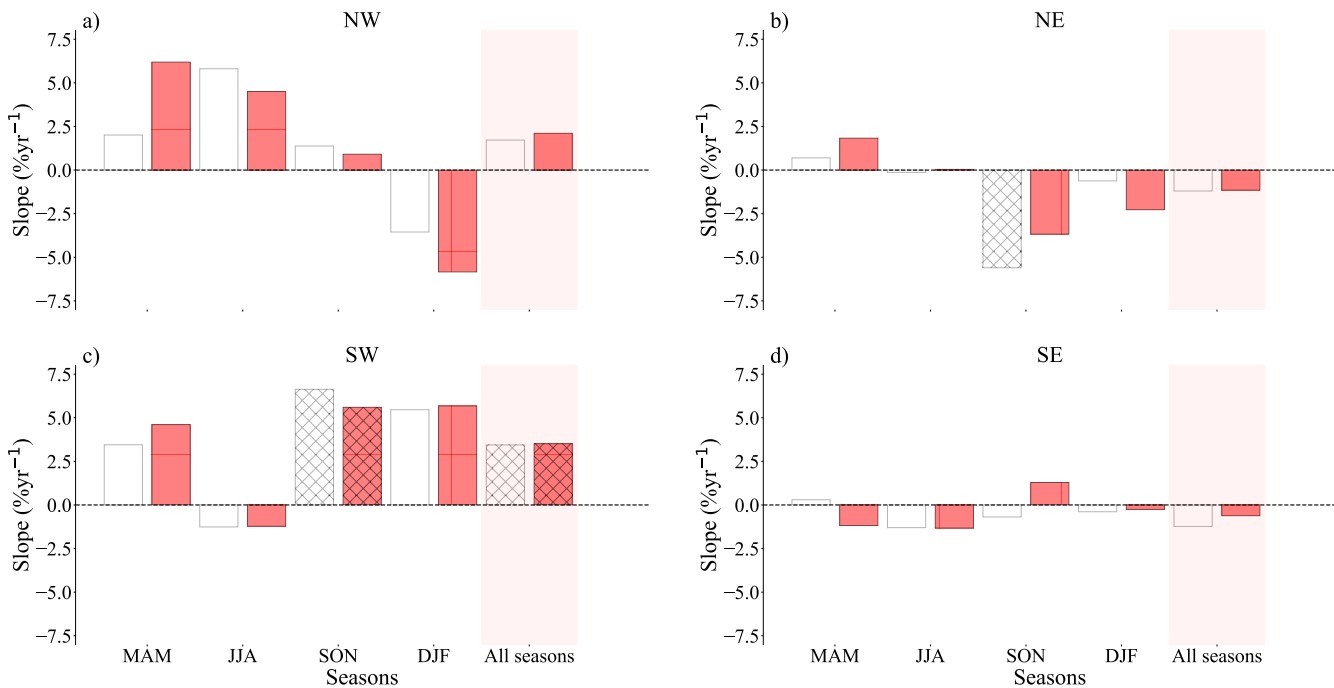

**Figure 5.** Relative trends in monthly contributions from each respective region: a) north-west (NW), b) north-east (NE), c) south-east (SE), and d) south-west (SW). The white bar displays the Theil-Sen estimator (TS), red bar displays log-transformed Least Mean Square (LMS) trends. Crosshatching denotes trends that are statistically significant at a confidence interval of 95 %.

The influence on $\alpha$ from the various surface types is displayed in the ternary plots (see Fig. 6). Each hexbin displays a unique combination of the different surface types namely, open water, land and ice (all within the ML). The hexbin near the top vertex of each ternary plot displays the median $\alpha$ observed at ZEP when the arriving back trajectories spend nearly all of their time above land. It is apparent that back trajectories that traverse over a relatively large proportion of land show higher $\alpha$ values (yellower). Descending the right edge (Land) from the top vertex, the individual hexbins represent back trajectories that spend an increasing amount of time over open water. It is evident that the median $\alpha$ is lower (i.e. bluer) as the relative amount of time over open water increases (hexbins near to the lower right vertices in Figs. 6). For Fig. 6a, the relative residence times are potentially skewed by back trajectories that spend very little time within the ML. There is an additional requirement on Fig. 6b to ensure that the back trajectories spend at least 40 % of the seven days within the ML. It is noticeable that in Fig. 6b that the number of data points is considerably lower for back trajectories that traversed mainly over land (see hexbins near to the top vertices in Fig. 6b), and thus do not meet the required minimum number. Svalbard is an archipelago and some considerable distance from the main continental landmasses, thus it makes sense for back trajectories that are completely dominated by land not to be present. Overall, there is a clear gradient in $\alpha$ values, as the % contribution from open water increases at the expense of land and ice. Moreover, this gradient is more pronounced as the amount of time back trajectories within the ML increases.





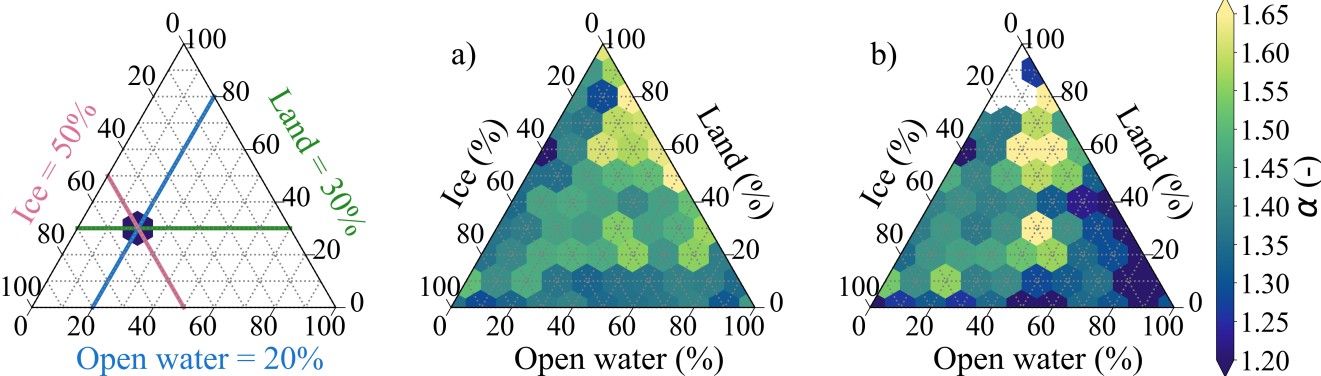

**Figure 6.** Scattering Ångström exponent ($\alpha$) as a function of surface type. Each hexbin displays the median of hourly $\alpha$ for that particular combination of surface types (see left schematic). The three vertices display the following surface contributions: 100% land (top), 100% open water (right), and 100% ice (left). The hourly $\alpha$ values are binned according to their proximity to the centre of a hexagon. The surface times are normalised using the time each back trajectory spends within the mixed-layer (ML). Certain criteria are placed on the back trajectories a) no criteria and b) 40 % of the 7 days were spent within the mixed layer. A minimum count of 50 has been placed on the hexbins.

Figure 7 correlates the seasonal $\alpha$ medians observed throughout the 18 years with the relative contribution of the combined north-west and south-west air masses. The correction is most striking for autumn with an r-value = - 0.59. Moreover, the influence from NW and SW air masses is largest in autumn. The variability in the contribution of NW and SW air masses for the different seasons is apparent; spring and winter show smaller contributions from NW and SW overall. Winter 2002 displays

an increased contribution from NW and SW air masses of ∼71%, providing a partial explanation for the seasonal low in $\alpha$.

## 4    Discussion

Table 1 presents an overview of the main findings. $\alpha$ decreases for all seasons, with a clear NW and SW increase throughout the year, and slight decreases in NE and SE in the autumn and winter. The seasonal trends in source regions all manifest, except for summer and winter, as trends in open water.

Increasing concentrations of coarse-mode particles relative to fine-mode particles reduce $\alpha$ (Schuster et al., 2006). Hence, the large decreasing trend in $\alpha$ (Fig. 1d) is suggestive of a significant shift to aerosol dominated more by coarse-mode particles. At the same time, increasing trends in $\sigma_{sp}$ and $\sigma_{bsp}$ are observed. Accumulation-mode particles are responsible for the majority of light scattering, however, as shown in Fig. S4 (in the supplementary material) at ZEP coarse-mode particles still contribute a significant amount to the overall light scattering. The relative increase in coarse-mode particles can, therefore, explain the

increasing trend in $\sigma_{sp}$ (Fig. 1a) as well. Smaller particles, on the other hand, exhibit increased values in $b$ (Seinfeld and Pandis, 2006). $b$ is more sensitive to the changes to the distribution of particles at the smaller end of the accumulation mode





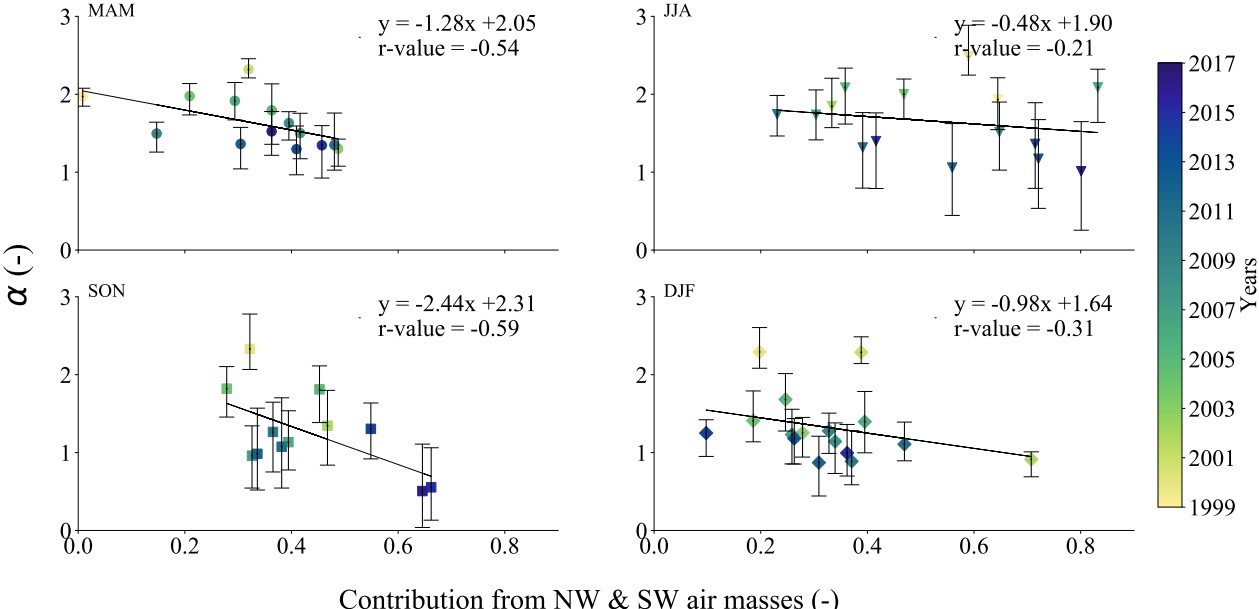

**Figure 7.** Relationship between the proportion of air masses arriving from the west (i.e. the combination of south-west and north-west regions) with seasonal scattering Ångström exponent ($\alpha$) medians. The four seasons namely spring (MAM), summer (JJA), autumn (SON), and winter (DJF) are presented. The error bars denote the length of the 25th and 75th percentile values. Note that the number of seasonal medians displayed is larger than the total number of distinct seasons, as winter here is subdivided into their respective years. The ordinary least squares method is applied to the seasonal medians.

(Collaud Coen et al., 2007). There is a small negative trend in $b$ (Fig. 4); nonetheless, a small relative decrease in $b$ could potentially mean substantial changes at the lower end of the accumulation-mode. The trends reported are consistent in sign and significance with those in Collaud Coen et al. (2020). It should be mentioned though that gaps in the data could affect the

calculated magnitude of the trends. Previous studies have used different averaging and "pre-whitening" methods to calculate the trends and their respective statistical significance (e.g., Pandolfi et al., 2018; Collaud Coen et al., 2013, 2020). However, by applying different statistical methods on both daily and seasonal averaging times, a better indication of the robustness of the derived trends is achieved.

     Sea spray aerosol (SSA) is the most plausible type of coarse-mode aerosol to be encountered at ZEP. It is produced at the

ocean surface through the process of wind-induced bubble-bursting and the tearing of wave crests (O'Dowd and De Leeuw, 2007). The production and transportation of SSA are linked to surface wind speeds (De Leeuw et al., 2011), with increasing sea-salt mass associated with increasing wind speeds (Lewis and Schwartz, 2004). SSA are identified by smaller $\alpha$ values, typically less than 0.5 (Schmeisser et al., 2017), and can contribute to the total amount of scattering. The clear oceanic influence on $\alpha$ (see Fig. 6), suggests that the relative influx of coarse-mode particles are marine in origin, and most likely SSA as opposed to

coarse-mode dominated dust from land sources.





**Table 1.** Schematic to aid the reader in the discussion related to the observed trends in aerosol light scattering properties and results from the trajectory analysis. Arrows correspond to the sign of the trends. Single arrows are given when the average of the two methods is greater than 1% or statistically significant. Double arrows display trends where the average trend is more than 2%. The asterisk corresponds to non-statistically significant results from the least squares method (LMS). The brackets correspond to non-statistically significant results from the Mann-Kendall (MK) test. All optical values are given for the 550 nm wavelength.

|  | Spring | Summer | Autumn | Winter | All seasons |
|---|---|---|---|---|---|
| $\sigma_{sp}$ | ↑ | ↑↑ | ↑↑ | ↑↑ | ↑↑ |
| $\alpha$ | ↓↓ | ↓↓ | ↓↓ | ↓↓ | ↓↓ |
| Open water | ↑↑ | (↓↓) | ↑↑ | - | ↑ |
| Precipitation | ↑↑ | - | ↑↑ | (↑)* | ↑↑ |
| Wind speed | - | - | (↑) | - | (↑) |
| NW | (↑)* | (↑↑)* | (↑)* | (↓↓)* | (↑)* |
| NE | (↑)* | - | ↓↓* | (↓)* | (↓)* |
| SE | - | (↓)* | - | - | - |
| SW | (↑↑)* | (↓)* | ↑↑ | (↑↑)* | ↑↑ |

Within this work, we show that the increased presence of SSA at ZEP is the result of changes in air circulation patterns, as opposed to the retreat in Arctic sea ice. There is a clear increase in the contribution of SW air masses, in particular in autumn, which coincides with the largest trend in $\alpha$. The trends in $\sigma_{sp}$ and $\alpha$ in summer can be explained by a greater influx of SSA arriving from the NW, over the Arctic sea ice, during the summer. Despite the continued increase in the expanse of open water

in the Arctic ocean, and the potential for increased SSA production, the effects of retreating sea ice on observations at ZEP were shown to be negligible (Fig. S5 in supplementary material). This study suggests that the growing contribution from western air masses leads to more SSA being transported to ZEP (Fig. 7). The overall effect is that air masses reaching ZEP are becoming more marine dominated. The largest increases in the expanse of summertime open water, within the Arctic Ocean, concern the Beaufort and Chukchi seas. These areas north of ZEP are a considerable distance from ZEP, thus the residence times back

trajectories spend over these regions are minimal compared to the combined area that all the back trajectories cover. However, more work is required to ascertain the exact annual movement of the Polar Front, and whether its location excludes ZEP from any changes in emissions induced by the melting summer sea ice.

The back trajectory analysis offers a plausible partial explanation of the changes in aerosol optical properties observed at ZEP. However, it should be noted that there is unlikely to be any single over-arching factor determining the observed trends.

Accompanying the changes in air circulation patterns, there is a positive trend in wind speeds along the back trajectories (Figs. 3 and 4). The inverse relationship between wind speeds and $\alpha$ (Woodcock, 1953), suggests that higher wind speeds allow larger particles to be produced and transported to the observatory.





The polar front advances in the winter to include more industrial emissions coming from Eurasia (as showed by the increased NE contribution in Fig. S6 in supplementary material). In the summertime, the polar front retreats and ZEP is found more often

outside of the polar front receiving mainly marine air masses from the Atlantic Ocean (Dall´Osto et al., 2017). The increased contribution from SW air masses in the autumn and winter (Fig. 4) over the 18 years, suggests that the time the polar front is residing north of ZEP is increasing; this further signals the observatory's shift from a polar to a marine site. Previous studies have shown that transport through the North Atlantic has become more frequent in the last decades (Mewes and Jacobi, 2019). The long-term increasing trend in the contribution from air masses from the SW in winter and NW in summer are consistent

with the findings in Maturilli and Kayser (2017), who have shown that in the winter troposphere, the frequency in wind from north-westerly directions is reduced, and instead, there is much more frequent wind from southern directions.

The overall changes in air circulation patterns can be tied to regional shifts in weather phenomena. It is well known that long-term trends in the North Atlantic Oscillation (NAO) influence long-term trends in Arctic pollution (Stohl, 2006). The Arctic Oscillation (AO) can also be used to describe the same phenomenon as the NAO. The AO index has been shown to be

shifted towards more negative anomalies in recent years (Maturilli and Kayser, 2017), which is consistent with the findings in this study. A negative AO increases the amount of meridional transport (south-to-north or vice versa) (Maturilli and Kayser, 2017); the Icelandic low pressure system and the Azores high pressure system are weakened, restricting westerly flow, and enabling an increased amount of meridional transport, over the North Atlantic.

Other potential explanations for the shift towards large particles exist. The following alternatives are addressed as to whether

these trends are the result of any other hypotheses: (1) increased influence of mineral dust in the Arctic; (2) decreases in new particle formation (NPF); (3) decreases in long-range transport and/or local emissions of anthropogenic pollution; (4) decreases in the scavenging of larger particles due to changes in the rates of wet deposition.

Mineral dust, arising from deglaciation in the Arctic, is another coarse-mode aerosol with the potential to be observed at ZEP. However, it is well established that SSA is the most abundant coarse-mode aerosol at ZEP (Weinbruch et al., 2012). The

clear open water dependency of $\alpha$ (see Fig. 6) suggests the changes are a result of marine influences, thus negating (1). Ultrafine particles ($< 100\,\text{nm}$) produced via NPF are considered to contribute a negligible amount to the total aerosol light scattering. Furthermore, NPF events at ZEP have been shown to be anti-correlated with sea ice extent (Dall´Osto et al., 2017), thus (2) is not considered. Anthropogenic pollution, which is either transported up to the Arctic or produced locally, is typically dominated by accumulation-mode aerosol particles and thus exhibits higher $\alpha$ values. If (3) is assumed and also that there is no increase

in the contribution from coarse-mode aerosol, it is difficult to explain the observed increasing trend in $\sigma_{\text{sp}}$. The measurements in this study, however, cannot be used to demonstrate a reduction in anthropogenic pollution at ZEP. However, previous studies have reported reductions in elemental black carbon (BC) at ZEP (Stone et al., 2014; Eleftheriadis et al., 2009; Hirdman et al., 2010), notable given the increase in BC emission inventories in the past decade (Ohara et al., 2007). Hirdman et al. (2010) also present reductions in measured total sulphate concentrations at ZEP. It should be noted that Hirdman et al. (2010) conclude

that about 4.9% and 0.3% of the reductions in BC and sulphate respectively at ZEP can be explained by circulation changes within the time period (1990–2009). Higher median $\alpha$ values for the air masses arriving from the north-east and south-east are observed (Fig. S7 in supplementary material). These air masses, where a considerable amount of Eurasian pollution originates



from, have decreased in their relative contribution. Wet deposition influencing Arctic aerosol requires a thorough analysis. However, the trend in precipitation from arriving back trajectories is increasing (Fig. 4c), as wetter air masses from the south-

west arrive at the observatory. The increase in precipitation suggests that (4) is unlikely to be the reason for the shift to more coarse-mode particles. Dall´Osto et al. (2017) show that NPF events are linked to the retreat in Arctic sea ice; ultrafine particles (< 100 nm) are shown to be associated with air masses travelling over open water and sea ice. In this study, SSA are linked to air masses traversing over open water, however, the overall mechanism that is ascribed to the observed aerosol trends is changes in air mass contributions.

## 5  Conclusion

Aerosol composition at ZEP has undergone significant changes over the last two decades, manifesting in a clear shift of aerosol light scattering properties. The statistically significant increasing trend in $\sigma_{\mathrm{sp}}$ of ∼2.4 to 2.9%yr$^{-1}$ combined with a statistically significant decreasing trend in $\alpha$ of around -4.9 to 6.3%yr$^{-1}$, demonstrates that aerosol observed at ZEP have become more dominated by coarse-mode particles over time, which in turn contribute a greater proportion to the scattering of light. There is

a clear open water dependency on $\alpha$, suggesting that the increasing trend is a result of marine influences leading to increased transport of SSA to the site. The strong decrease of Arctic sea ice over the last decades, leading to more open water, is, however, not the main reason for the increased contribution of SSA. Here, we demonstrate, that the growing marine influence is originating from an increasing time back trajectories spend over the open water particularly south-west from Svalbard. As such, the changes in air circulation patterns have resulted in a characteristic shift of ZEP, from an Arctic towards a more marine

dominated site.

ZEP is a site that requires detailed analysis, given the significance and magnitude of the observed trends. The results in this study suggest that climate-related changes are influencing the transportation of aerosol particles in and to the Arctic region, as well as the processing and sources of particles. It is important to note that the results are site dependent and that no general conclusions for the entire Arctic can be made.

*Data availability.*  The data of this study will be available on the Bolin Centre Database (DOI and link will be added later).

*Author contributions.*  DHR performed data analysis and wrote manuscript together with PZ with input from all co-authors. PT performed trajectory calculations. HCH and JS were part of the startup and operation of the long-term measurements. All authors read and commented on the manuscript.

*Competing interests.*  PZ and RK are acting as co-editors with ACP. No further competing interests are applicable.





*Acknowledgements.* We would like to thank research engineers Birgitta Noone, Tabea Henning, and Ondrej Tesar from ACES and the staff from the Norwegian Polar Institute (NPI) for their on-site support. NPI is also acknowledged for substantial long-term support in maintaining the measurements at Zeppelin Observatory. This work was financially supported by the long-term support of the Swedish EPA's (Naturvårdsverket) Environmental monitoring program (Miljöövervakning) and the Knut-and-Alice-Wallenberg Foundation within the ACAS project (Arctic Climate Across Scales, project no. 2016.0024). We thank the Norwegian Institute for Air Research (NILU) for

providing the ambient meteorological data and for maintaining the EBAS database. We thank Elisabeth Andrews (NOAA/CIRES, U.S.A.) for support and guidance.



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
