# Peer review of "From a polar to a marine environment: has the changing Arctic led to a shift in aerosol light scattering properties?"

_Atmospheric Chemistry and Physics, 2020_

## Referee Comment (RC1) · Anonymous Referee #1 · 14 Jul 2020

Summary:

The analysis outlined in this manuscript utilizes long-term (17 year) aerosol measurements from Zeppelin observatory in Svalbard, Norway to explore if observed climate changes in the Arctic are apparent in characteristics of the aerosol population. Trends in aerosol light scattering, backscattering, scattering Ångström exponent, and hemispheric backscattering fraction are computed. The authors find a statistically significant increase in aerosol light scattering coefficient at wavelength 550nm and a decrease in scattering Ångström exponent at wavelengths 450 and 550nm, indicating a shift to more coarse-mode aerosol. The conclusion is that the observatory is measuring more

coarse mode aerosol, sea salt in particular, due to shifts in winds bringing more air masses from the southwest (as opposed to influence of melting sea ice).

General Comments:

This manuscript presents an important scientific analysis of aerosols at one Arctic monitoring station, the results of which are well within the scope of ACP. The quality of the scientific methods are clear- the approach is methodical and thorough, exploring multiple physical mechanisms that could explain the trends in aerosol data. Results and supporting evidence are convincing and communicated efficiently. The manuscript is very well written; I found very few technical corrections.

Specific Comments:

In abstract: The sentence "The scattering Ångström exponent and the particle light scattering coefficient exhibit statistically significant decreasing of between -4.9 and -6.3 % per year (using wavelengths of $\lambda$ = 450 and 550 nm) and increasing trends of between 2.3 and 2.9 % per year (at a wavelength of $\lambda$ = 550 nm), respectively" is easy to misinterpret. It took much too long to decipher what was being communicated. Considering clarifying the sentence with a simple change like this: "The scattering Ångström exponent exhibits statistically significant decreasing of between -4.9 and -6.3 % per year (using wavelengths of $\lambda$ = 450 and 550 nm), while the particle light scattering coefficient exhibits statistically significant increasing trends of between 2.3 and 2.9 % per year (at a wavelength of $\lambda$ = 550 nm)."

In methods section: Please include temporal resolution of the sampling from the nephelometer. Page 5, Line 150 mentions that 5 data points are used to compute hourly medians, but it is not clear what percentage of the total hourly data points that is.

Page 6, Line 166: Is there a large diurnal cycle in aerosol properties at ZEP? When computing long-term trends, is it important that that diurnal cycle is obscured by using daily medians? In other words, do you have any reason to suspect the long-term trends

in daytime vs. nighttime (or maxima vs. minima) aerosol properties look different?

Figure 2 (& Figure 4): What is the pink bar surrounding the 'all seasons' bars? If it is just to set apart the all seasons from the other seasons, it is a bit misleading on the bar plot because it looks like it is a bar representing data- I looked for a legend or explanation for the pink bars for a while. Maybe just a black line separating the 'all seasons' from the other seasons would be sufficient. Or simply specifying what the pink shading is in the caption would be helpful for the reader.

Page 11, Line 277: What happens if a back trajectory crosses multiple specified regions (SE, SW, NW, NE), as I'm sure happens quite often? How is it classified? Is it classified by where the back trajectory originated, or by the sector from which it directly approached the station immediately before arrival? It might be good to clarify this in the methods section when discussing the back trajectory region definitions.

Technical Corrections:

Page 2, Line 33: 'report' should be 'reports' (since the Panel is singular)

Page 3, Line 71: add 'respectively' after 'wavelengths $\lambda 1$ and $\lambda 2$.'

Page 5, Line 140: remove ',' after 'Approximately'

Page 6, Line 157: 'studies often use a constant' threshold

Page 6, Line 158: add units after $\sigma sp > 1$ (Mm-1)

Page 7, Line 204: remove '-' after (Jones et al., 2001)

Page 13, Line 310: Remove the first 'that' in 'It is noticeable that in Fig. 6b that the'

---

## Referee Comment (RC2) · Anonymous Referee #2 · 7 Aug 2020

GENERAL

The paper presents an 18-yr data record of aerosol optical properties measured at the Zeppelin observatory in Svalbard. The optical properties discussed in the paper are total scattering coefficient, backscattering coefficient, their ratio and the wavelength dependency of scattering. The data are analysed especially to find trends and to interpret these. The trends are analysed using statistical methods that yield more or less similar results. For the interpretation also transport analyses were conducted using the HYSPLIT model. The observed trend is obviously towards more marine aerosol. The authors show that show that changes in air mass circulation patterns are the main

factor responsible for the trend, not the decrease of Arctic sea ice. This is an important result and should be published.

The only thing that slightly puzzles me is year 2001. Fig 1 shows that both the backscatter fractions and scattering Ångström exponents are clearly higher then than before or after it, it looks like an outlier. What is special in 2001? Is there any possibility of a technical explanation? Inlet issue, instrument issue or similar? Or forest fires from Siberia? They emit small particles. If you omitted 2001, how would the trends, their statistical significance, and conclusions look like?

Answering that and the small editing suggestions – not requirements – I present below are enough for publishing the paper in ACP.

DETAILED COMMENTS

Lines 57-73 present equations that are somewhat an outlier in the introduction. Those lines would much more logically belong to section 2.4. Think about moving them. I don't require that, though, but the move would make the introduction more fluent.

Table 1. In the caption it is written " All optical values are given for the 550 nm wavelength." But there are only two optical properties, scattering coefficient and Ångström exponent and Ångström exponent is not at 550 nm. Reword.

Tables in the supplement Table S1, in the caption it is written "Decreasing (D) and increasing (I) statistically significant trends are signified." But there are no decreasing trends in the table. Consider rewording. Analogous comment for Tables S2-S4.

---

## Author Response (AR1)

**Reply to reviewers of the manuscript "From a polar to a marine environment: has the changing Arctic led to a shift in aerosol light scattering properties?"**

Heslin-Rees et al.

September 23, 2020

We thank both reviewers for their positive and constructive comments. We have modified our manuscript based on their suggestions. Please find our detailed reply below (given in blue colour).

**1   Reviewer 1**

The analysis outlined in this manuscript utilizes long-term (17 year) aerosol measurements from Zeppelin observatory in Svalbard, Norway to explore if observed climate changes in the Arctic are apparent in characteristics of the aerosol population. Trends in aerosol light scattering, backscattering, scattering Ångström exponent, and hemispheric backscattering fraction are computed. The authors find a statistically significant increase in aerosol light scattering coefficient at wavelength 550nm and a decrease in scattering Ångström exponent at wavelengths 450 and 550nm, indicating a shift to more coarse-mode aerosol. The conclusion is that the observatory is measuring more coarse mode aerosol, sea salt in particular, due to shifts in winds bringing more air masses from the southwest (as opposed to influence of melting sea ice). This manuscript presents an important scientific analysis of aerosols at one Arctic monitoring station, the results of which are well within the scope of ACP. The quality of the scientific methods are clear the approach is methodical and thorough, exploring multiple physical mechanisms that could explain the trends in aerosol data. Results and supporting evidence are convincing and communicated efficiently. The manuscript is very well written; I found very few technical corrections.

2. In abstract:
The sentence "The scattering Ångström exponent and the particle light scattering coefficient exhibit statistically significant decreasing of between -4.9 and -6.3 % per year (using wavelengths of $\lambda = 450$ and $550\,\text{nm}$) and increasing trends of between 2.3 and 2.9 % per year (at a wavelength of $\lambda = 550\,\text{nm}$), respectively." is easy to misinterpret. It took much too long to decipher what was being communicated. Considering clarifying the sentence with a simple change like this: "The scattering Ångström exponent exhibits statistically significant decreasing of between -4.9 and -6.3 % per year (using wavelengths of $\lambda = 450$ and $550\,\text{nm}$), while the particle light scattering coefficient exhibits statistically significant increasing trends of between 2.3 and 2.9 % per year (at a wavelength of $\lambda = 550\,\text{nm}$)"

We agree that the sentence in the abstract could be misinterpreted, and it is simpler to separate out the two findings. We have changed the sentence in the abstract to the one suggested

in the comment: "The scattering Ångström exponent exhibits statistically significant decreasing of between -4.9 and -6.3 % per year (using wavelengths of $\lambda = 450$ and $550\,\mathrm{nm}$), while the particle light scattering coefficient exhibits statistically significant increasing trends of between 2.3 and 2.9 % per year (at a wavelength of $\lambda = 550\,\mathrm{nm}$)."

3. In the methods section:
Please include temporal resolution of the sampling from the nephelometer. Page 5, Line 150 mentions that 5 data points are used to compute hourly medians, but it is not clear what percentage of the total hourly data points that is.

The use of 5 data points is incorrect as the nephelometer conducts continuous averaging (boxcar) itself, before logging the data. The averaging period does change. However, during post-processing, the short averaged data can be combined with the longer averages (TSI Incorporated, 2005). All logged data points were therefore considered valid and no minimum number of points per hour was required. The data was therefore re-analysed without the 5 data point threshold imposed. Removing the 5 data point threshold meant that there was slightly more data available to use, as a result, the trends have changed very slightly in light of the extra data points. The proportion of hourly data points used in this study, compared with the number of raw (did not undergo quality control procedure) hourly data points, is displayed in figure 1. This figure has been added to the supplementary material (replaces previous figures S1).

[Figure]

Figure 1: The number of hourly data points prior to (black bar) and post (blue bar) quality control procedures. Notice that the number of hourly data points for the light scattering coefficients ($\lambda = 450\,\text{nm}$ and $\lambda = 550\,\text{nm}$) and backscattering coefficient ($\lambda = 550\,\text{nm}$) are less than the blue bar. The bar blue represents the number of hourly averages of either one of these three main variables.

Page 6, Line 166:
Is there a large diurnal cycle in aerosol properties at ZEP? When computing long-term trends, is it important that that diurnal cycle is obscured by using daily medians? In other words, do you have any reason to suspect the long-term trends in daytime vs. nighttime (or maxima vs. minima) aerosol properties look different?

The diurnal cycle for the main aerosol optical properties at ZEP has been explored on the recommendation of this comment. No obvious diurnal cycles were noticed for both the scattering Ångström exponent and the particle light scattering coefficient (see Figs 2 and 3), however, the summer months exhibit slightly more variability. The amount of available sunlight during the summer months (i.e. the polar day) allows for more photochemical processes, and thus leads to more new particle formation (NPF). Given the limitations on the ability of the nephelometer to detect particles that arise from NPF (i.e., particles are too small for scattering visible light), it is assumed that any diurnal processes present in the light scattering properties are not the result of NPF events. The absence of pronounced daily cycles is not surprising for this location. Advection and transportation of local aerosol particles is influenced by diurnal variation. However, the remoteness and altitude of the Zeppelin Observatory, which is located on a mountain with less local meteorological influence, means that little anthropogenic aerosol sources are affected by these diurnal cycles of advection and transportation. For example, sites in Barrow and Atqasuk report a weak aerosol diurnal variation for the aerosol optical depth and Ångström exponent (Yin and Min, 2014). The trends corresponding to only daylight and nighttime observations were further examined (see Figs 4 and 5).

The following sentence was added to the manuscript on line 169:

[Figure]

Figure 2: The diurnal cycles for the scattering Ångström exponent ($\alpha$). The local standard time is displayed as 24 hours. The seasonal medians are denoted by their respective symbols. The error bars denote the length of the 25th and 75th percentile values. The seasonal mean is given by the cross.

"It should be noted that the light scattering properties at ZEP do not show any pronounced daily cycle (not shown)."

[Figure]

Figure 3: The diurnal cycles for light scattering coefficient ($\sigma_{\mathrm{sp}}$, at a wavelength of $\lambda = 550\,\mathrm{nm}$). The local standard time is displayed as 24 hours. The seasonal medians are denoted by their respective symbols. The error bars denote the length of the 25th and 75th percentile values. The seasonal mean is given by the cross.

Figure 2 (& Figure 4):

What is the pink bar surrounding the 'all seasons' bars? If it is just to set apart the all seasons from the other seasons, it is a bit misleading on the bar plot because it looks like it is a bar representing data- I looked for a legend or explanation for the pink bars for a while. Maybe just a black line separating the 'all seasons' from the other seasons would be sufficient. Or simply specifying what the pink shading is in the caption would be helpful for the reader.

The pink shaded area was probably misleading (it had no special meaning). The figures (see Figs. 6-8) have been altered to include a dashed line instead of the shaded region. An additional explanation has been added to the caption.

[Figure]

Figure 4: Trends for daylight observations: Long-term trends of the seasonal medians for a) the particle light scattering coefficient ($\lambda = 550\,\mathrm{nm}$) b) the particle light backscattering coefficient ($\lambda = 550\,\mathrm{nm}$) c) the hemispheric backscattering fraction ($\lambda = 550\,\mathrm{nm}$) d) the scattering Ångström exponent ($\lambda_1 = 450\,\mathrm{nm}$, $\lambda_2 = 550\,\mathrm{nm}$). The seasonal medians are denoted by their respective symbols. The error bars denote the length of the 25th and 75th percentile values. The seasonal mean is given by the cross. The solid and dashed red lines represent the least mean square (LMS) and Theil-Sen slope (TS) of the seasonal medians, respectively. The red shaded area denotes the associated 90 % confidence interval of the TS slope. Note that TS is not used to test the statistical significance.

[Figure]

Figure 5: Trends for nighttime observations: Long-term trends of the seasonal medians for a) the particle light scattering coefficient ($\lambda = 550\,\mathrm{nm}$) b) the particle light backscattering coefficient ($\lambda = 550\,\mathrm{nm}$) c) the hemispheric backscattering fraction ($\lambda = 550\,\mathrm{nm}$) d) the scattering Ångström exponent ($\lambda_1 = 450\,\mathrm{nm}$, $\lambda_2 = 550\,\mathrm{nm}$). The seasonal medians are denoted by their respective symbols. The error bars denote the length of the 25th and 75th percentile values. The seasonal mean is given by the cross. The solid and dashed red lines represent the least mean square (LMS) and Theil-Sen slope (TS) of the seasonal medians, respectively. The red shaded area denotes the associated 90 % confidence interval of the TS slope. Note that TS is not used to test the statistical significance.

[Figure]

Figure 6: Relative trends based on daily medians for a) particle light scattering coefficient, b) particle light backscattering coefficient, c) hemispheric backscattering fraction (note the different y-scale), and d) scattering Ångström exponent, for different and all seasons. The white bar displays the Theil-Sen estimator (TS). The red bar displays log-transformed Least Mean Square (LMS) trends. Crosshatching denotes trends that are statistically significant (ss) at a confidence interval of 95 %. The ss for the TS is based on "prewhitened" (PW) time series. The trends in their respective units $\mathrm{yr}^{-1}$ are in the tables in the appendix. The dashed line aids the reader in separating the individual seasonal trends and the trend with all the seasons included.

[Figure]

Figure 7: Relative trends in daily medians for a) time spent above open water and within the ML, b) median wind speed, c) accumulated precipitation along each back trajectory for different and all seasons. The white bar displays the Theil-Sen estimator (TS). The red bar displays log-transformed Least Mean Square (LMS) trends. Crosshatching denotes trends that are statistically significant (ss) at a confidence interval of 95%. The ss for the TS is based on "prewhitened" (PW) time series. The dashed line aids the reader in separating the individual seasonal trends and the trend with all the seasons included.

[Figure]

Figure 8: Relative trends in monthly contributions from each respective region: a) north-west (NW), b) north-east (NE), c) south-east (SE), and d) south-west (SW). The white bar displays the Theil-Sen estimator (TS), red bar displays log-transformed Least Mean Square (LMS) trends. Crosshatching denotes trends that are statistically significant at a confidence interval of 95 %. The dashed line aids the reader in separating the individual seasonal trends and the trend with all the seasons included.

Page 11, Line 277:
What happens if a back trajectory crosses multiple specified regions (SE, SW, NW, NE), as I'm sure happens quite often? How is it classified? Is it classified by where the back trajectory originated, or by the sector from which it directly approached the station immediately before arrival? It might be good to clarify this in the methods section when discussing the back trajectory region definitions.

The back trajectories do cross multiple regions, however, it is assumed that by taking the average of the coordinates along each back trajectory, and generating a mean coordinate (mean latitude, mean longitude), that coordinate will be somewhat representative of the direction in which the air parcels have travelled from. Lines 194 - 196 in the manuscript, and Figure S3 (in the supplement) provide some explanation. However, additional clarification has been added to the manuscript to help explain this possible confusion. The extra sentence was added to section 2.4.3. Trajectory calculations, line 195:

"[The direction from which back trajectories arrive at ZEP is computed by calculating the mean Cartesian-transformed coordinates]. The mean coordinate is used to assign each back trajectory a region namely, north-west, north-east, south-east, and south-west. The region the back trajectories are assigned is dependent on the average of the coordinates, so not defined based on the origin. It is understood that back trajectories can cross multiple regions, the assigned regions simplify the classification."

Technical Corrections:

Page 2, Line 33:
'report' should be 'reports' (since the Panel is singular)
We agree and have made the following correction:
"The IPCC (2013) reports that in combination with clouds, aerosols continue to contribute the largest uncertainty to our understanding of changes to the Earth's energy budget."

Page 3, Line 71:
add 'respectively' after 'wavelengths $\lambda_1$ and $\lambda_2$'
We agree and have made the following correction:
"where $\sigma_{\mathrm{sp},1}$ and $\sigma_{\mathrm{sp},2}$ are the particle light scattering coefficients at wavelengths $\lambda_1$ and $\lambda_2$ respectively."

Page 5, Line 140:
remove ',' after 'Approximately'
We agree and have made the following correction:
"Approximately $\sim$59.2 % of the hourly aerosol measurements are left in after the quality control procedure and temporal collocation of the data set."

Page 6, Line 157:
'studies often use a constant' threshold
We agree and have made the following correction:
"Small values for $\sigma_{\mathrm{sp}}$ are considered less reliable due to instrument noise at low aerosol loadings (Schmeisser et al., 2017), and studies often use a constant threshold, Schmeisser et al. (e.g., 2018) consider $\sigma_{\mathrm{sp}} > 1$"

Page 6, Line 158:
add units after $\sigma_{\mathrm{sp}} > 1$ $(\mathrm{Mm}^{-1})$
"Small values for $\sigma_{\mathrm{sp}}$ are considered less reliable due to instrument noise at low aerosol loadings

(Schmeisser et al., 2017), and studies often use a constant threshold, Schmeisser et al. (e.g., 2018) consider $\sigma_{\mathrm{sp}} > 1$ $(\mathrm{Mm}^{-1})$”

Page 7, Line 204:
remove '-' after (Jones et al., 2001)
We agree and have made the following correction:
“For all the trend analyses, the Python *scipy.stats* package is used within *SciPy* (v.1.1.0) (Jones et al., 2001).”

Page 13, Line 310:
Remove the first 'that' in 'It is noticeable that in Fig. 6b that the'
We agree and have made the following correction:
“It is noticeable in Fig. 6b that the number of data points is considerably lower for back trajectories that traversed mainly over land (see hexbins near to the top vertices in Fig. 6b), and thus do not meet the required minimum number.”

**2 Reviewer 2**

The paper presents an 18-yr data record of aerosol optical properties measured at the Zeppelin observatory in Svalbard. The optical properties discussed in the paper are total scattering coefficient, backscattering coefficient, their ratio and the wavelength dependency of scattering. The data are analysed especially to find trends and to interpret these. The trends are analysed using statistical methods that yield more or less similar results. For the interpretation also transport analyses were conducted using the HYSPLIT model. The observed trend is obviously towards more marine aerosol. The authors show that show that changes in air mass circulation patterns are the main factor responsible for the trend, not the decrease of Arctic sea ice. This is an important result and should be published. The only thing that slightly puzzles me is year 2001. Fig 1 shows that both the backscatter fractions and scattering Ångström exponents are clearly higher then than before or after it, it looks like an outlier. What is special in 2001? Is there any possibility of a technical explanation? Inlet issue, instrument issue or similar? Or forest fires from Siberia? They emit small particles. If you omitted 2001, how would the trends, their statistical significance, and conclusions look like? Answering that and the small editing suggestions – not requirements – I present below are enough for publishing the paper in ACP.

Despite the reviewer mentioning 2001 in the comments above, we expect that they are instead referring to 2002/2003 (see Fig. 1a).
The start of 2002 and the beginning of 2003, corresponding to winter and spring (during the Arctic Haze period), display seasonal medians significantly different from both neighbouring years. The light scattering coefficients are much larger for these particular seasons, and the scattering Ångström exponent is somewhat smaller. These seasons definitely represent outliers. To the best of our knowledge there is no technical explanation for this. In terms of instrument maintenance, the nephelometer was sent for repairs at the end of 2003, and returned sometime in February/March 2004 (hence the gap in data during this period). Figures 9 and 10 present the measurements recorded by the nepheleometer prior to any quality control procedures. The period before the nepheleometer was sent away for repairs in 2003 is displayed (see Fig. 10). We think that the data represented in the figures in the manuscript are valid, despite these particular seasons displaying somewhat different medians. The increased contribution from both north-west and south-west air masses could help to explain the anomalous results in the winter

and spring of 2002-03 (see lines 319 - 320 of the manuscript).

To support the claim that these data points are indeed valid, we point to examples in the literature, and also to other aerosol instrumentation that was operated at the Zeppelin Observatory during this period.

The nephelometer measurements are also consistent with other independent observations from the same period. Calculated particle light scattering coefficients using Mie theory and particle size distributions (DMPS) see a similar increase in 2003 as the nephelometer (see Fig 11). The Mie derived scattering coefficient, in winter 2002/03, matches well with the reported nephelometer observations recorded for that season. The overall underestimation of the calculated values is most likely due to the fact that the DMPS only measured until 950 nm and the assumption of a constant refractive index. This suggests that other instrumentation, other than the nepheleometer, experienced elevated concentrations and/or larger aerosol particles.

In addition, this increase has also been observed in the literature by means of sun photometry and satellite observations ((Glantz et al., 2014; Eleftheriadis et al., 2009a; Myhre et al., 2006)). There is a distinct increase in the aerosol optical depth (AOD) around Svalbard during the spring of 2003, in particular, May, where the median daily AOD reached 0.3 (Glantz et al., 2014). Moreover, (Myhre et al., 2006) demonstrate that there are periods in which AOD values are elevated. AOD measurements show a high daily median towards the end of March 2003 (Myhre et al., 2006). Furthermore, Eleftheriadis et al. (2009b) show a distinct increase in BC concentrations for 2003 as well.

Given that the peaks in the light scattering coefficient appear in winter and spring, it is unlikely to be the result of forest fires. Aerosol particles from biomass burning, which are then transported to Svalbard via long-range transport, influence the aerosol optical properties during the summer months (Glantz et al., 2014); one notable example of such a case was the Canadian forest fires of July 2004 (Stohl, 2006).

[Figure]

Figure 9: Timeseries of raw nepheleometer data for the year 2002, prior to any cleaning. NBXX, presented in black, is the nepheleometer data in normal measurement mode in which the total scattering and backscattering is recorded. ZBXX, in blue, demonstrates measurements in the zero mode. BBXX, in green, presents the measurements in the blanking mode. The normal mode is when the Rayleigh scattering signal is subtracted to give the scattering coefficient, whereas in zero mode it is not. In blanking mode the scattering coefficients retain their value from the previous mode.

[Figure]

Figure 10: Timeseries of raw nepheleometer data for the year 2002, prior to any cleaning. NBXX, presented in black, is the nepheleometer data in normal measurement mode in which the total scattering and backscattering is recorded. ZBXX, in blue, demonstrates measurements in the zero mode. BBXX, in green, presents the measurements in the blanking mode. The normal mode is when the Rayleigh scattering signal is subtracted to give the scattering coefficient, whereas in zero mode it is not. In blanking mode the scattering coefficients retain their value from the previous mode. The shaded region denotes the data which was removed in the cleaning process - notice that there was no blanking or zeroing (i.e. calibration) taking place, so that is why the data was removed, and ultimately why it was sent for repairs.

[Figure]

Figure 11: Mie-calculated light scattering coefficients ($\sigma_{\text{sp}}$, at a wavelength of $\lambda = 550\,\text{nm}$) is compared with the observational scattering coefficients from the nepheleometer. The Mie-scattering is calculated based on the assumption of a refractive index of m = 1.544+0j (Sodium chloride). Seasonal medians are denoted as dots, whilst the seasonal means are given as crosses. The seasons are defined based on the calendar (DJF, MAM, JJA, SON). Increased values of light scattering (e.g., particle surface) in 2002-2003 are observed by both instruments.

The year 2003 has been removed from the trend line, there is a change in the trend lines (based on seasonal medians). The influence is as follows:

[Figure]

Figure 12: Data from the year 2003 removed: Long-term trends of the seasonal medians for a) the particle light scattering coefficient ($\lambda = 550$ nm) b) the particle light backscattering coefficient ($\lambda = 550$ nm) c) the hemispheric backscattering fraction ($\lambda = 550$ nm) d) the scattering Ångström exponent ($\lambda_1 = 450$ nm, $\lambda_2 = 550$ nm). The seasonal medians are denoted by their respective symbols. The error bars denote the length of the 25th and 75th percentile values. The seasonal mean is given by the cross. The solid and dashed red lines represent the least mean square (LMS) and Theil-Sen slope (TS) of the seasonal medians, respectively. The red shaded area denotes the associated 90 % confidence interval of the TS slope. Note that TS is not used to test the statistical significance.

DETAILED COMMENTS:

Lines 57-73:
Present equations that are somewhat an outlier in the introduction. Those lines would much more logically belong to section 2.4. Think about moving them. I don't require that, though, but the move would make the introduction more fluent.
We agree and have included the equations in section 2.4.

Table 1.:
In the caption it is written " All optical values are given for the 550 nm wavelength." But there are only two optical properties, scattering coefficient and Ångström exponent and Ångström exponent is not at 550 nm. Reword.
The scattering Ångström exponent uses wavelengths of ($\lambda = 450$ and 550 nm), while the particle light scattering coefficient is given for the 550 nm wavelength.

Tables in the supplement Table S1:
In the caption it is written "Decreasing (D) and increasing (I) statistically significant trends are signified." But there are no decreasing trends in the table. Consider rewording. Analogous comment for Tables S2-S4.
The captions on the tables in the supplement have been changed according to the comments.

**3 Further changes**

Seasons can be defined in different ways, especially when it comes to Arctic aerosol measurements. Often, the seasons are separated into the "Arctic Haze" period (typically occurring in late winter and early spring) and the summer; the idea in these cases is to distinguish between two very contrasting periods when it comes to anthropogenic influence in the Arctic. However, in this study the seasons are defined by their respective calendar months.

The following sentence has been added to line 166 in order add extra clarification: "Daily and seasonal medians are computed and used to assess the trends in aerosol optical properties. The seasons are defined based on calendar dates; winter (December - February), spring (March - May), summer (June - August) and autumn (September - November)".

Slight changes to some of the stated values were made. The following lines have been altered in respect to the extra data:

Line 11 - 13:
"The scattering Ångström exponent exhibits statistically significant decreasing of between - 4.9 and -6.5 % per year (using wavelengths of $\lambda = 450$ and $550\,\text{nm}$), while the particle light scattering coefficient exhibits statistically significant increasing trends of between 2.6 and 2.9 % per year (at a wavelength of $\lambda = 550\,\text{nm}$)."

Line 140 -141:
"After the temporal collocation of the data set, approximately $\sim$52.8 % and 60.4% of the quality-controlled hourly medians for $\sigma_{\text{sp}}$ and $\sigma_{\text{bsp}}$ ($\lambda = 550\,\text{nm}$) are left the data set respectively."

Line 152 - 153:
"Hourly medians are calculated (see Fig. 1 in the supplement)."

Line 159 - 162:
"Overall, the fraction of data removed in terms hourly averages are as follows: 31.1% for $\sigma_{\text{sp}}$ ($\lambda = 450\,\text{nm}$), 26.1% $\sigma_{\text{sp}}$ ($\lambda = 550\,\text{nm}$) and 39.4% $\sigma_{\text{bsp}}$ ($\lambda = 550\,\text{nm}$). Most of the years are not affected by missing data with the exception of the years 2003 and 2016, where 66.6% -67.0% and 67.9% -74.0% of data is excluded respectively."

Line 179 - 182:
"The observatory has frequent inside-cloud situations, which can affect the aerosol optical measurements; approximately 10.9 % of the optical data is removed as a result of high ambient RH values, with summer the most affected season, $\sim 21.0$ % is removed, as opposed to $\sim 5.9$ %, $\sim 12.1$ %, $\sim 6.8$ % for spring, autumn, and winter, respectively."

Line 258 - 265:
Minor changes were made to certain stated values.

Line 274:
$\sim 0.62\,\text{ms}^{-1}$

Line 281:
air mass contributions (i.e. $\sim 37$ % and 33 %, respectively)

Line 287:
The time spent over open water and within the ML displays large statistically significant trends, in particular, spring and autumn show large positive relative changes ($3.3\,\%\mathrm{yr}^{-1}$ and $3.4\text{-}3.5\%\mathrm{yr}^{-1}$ respectively)

Line 294:
The south-west displays statistically significant increasing trends for autumn and across all seasons ($6.0\text{-}6.7\%\mathrm{yr}^{-1}$ and $2.6\text{-}3.2\%\mathrm{yr}^{-1}$ respectively).

The following sentence was added to line 250:
Furthermore, Eleftheriadis et al. (2009b) show increased BC concentrations for 2003.

[revised manuscript text omitted]